# Rearing in an Enriched Environment Ameliorates the ADHD-like Behaviors of Lister Hooded Rats While Suppressing Neuronal Activities in the Medial Prefrontal Cortex

**DOI:** 10.3390/cells11223649

**Published:** 2022-11-17

**Authors:** Ryo Utsunomiya, Kanta Mikami, Tomomi Doi, Mohammed E. Choudhury, Toshihiro Jogamoto, Naohito Tokunaga, Eiichi Ishii, Mariko Eguchi, Hajime Yano, Junya Tanaka

**Affiliations:** 1Department of Molecular and Cellular Physiology, Graduate School of Medicine, Ehime University, Toon 791-0295, Japan; 2Department of Pediatrics, Graduate School of Medicine, Ehime University, Toon 791-0295, Japan; 3Division of Medical Research Support, the Advanced Research Support Center (ADRES), Ehime University, Toon 791-0295, Japan; 4Department of Pediatrics, Imabari City Medical Association General Hospital, Imabari 794-0026, Japan

**Keywords:** LHR, enriched environment, prefrontal cortex, RNAseq, immediate-early gene, cFos

## Abstract

In addition to genetic factors, environmental factors play a role in the pathogenesis of attention deficit/hyperactivity disorder (ADHD). This study used Lister hooded rats (LHRs) as ADHD model animals to evaluate the effects of environmental factors. Male LHR pups were kept in four rearing conditions from postnatal day 23 (4 rats in a standard cage; 12 rats in a large flat cage; and 4 or 12 rats in an enriched environment [EE]) until 9 weeks of age. EE rearing but not rearing in a large flat cage decreased the activity of LHRs in the open field test that was conducted for 7 consecutive days. In the drop test, most rats reared in an EE remained on a disk at a height, whereas most rats reared in a standard cage fell off. RNA sequencing revealed that the immediate-early gene expression in the medial prefrontal cortex of LHRs reared in an EE was reduced. cFos-expressing neurons were reduced in number in LHRs reared in an EE. These results suggest that growing in an EE improves ADHD-like behaviors and that said improvement is due to the suppression of neuronal activity in the mPFC.

## 1. Introduction

Attention deficit/hyperactivity disorder (ADHD) is a common behavioral and neurodevelopmental disorder (NDD) characterized by hyperactivity, inattention, and impulsivity. It affects several percentages of children and often persists into adulthood [1,2,3]. Adults with ADHD are said to experience various difficulties, including drug abuse, depression, accidents, and domestic violence. Methylphenidate, a psychostimulant, and nonpsychostimulants, such as atomoxetine, are widely used as highly effective treatments for ADHD [4].

Many reports have described genetic influences on the development of ADHD; particularly, mutations in genes affecting monoaminergic neurotransmitters or fluctuations in expression have been reported as critical causes of ADHD [4,5]. Dopamine transporter knockout mice and spontaneously hypertensive rats (SHRs) have often been used as ADHD models [6,7,8,9] that are indicative of the critical role of genetic factors in the pathogenesis of ADHD. We recently reported that outbred Lister hooded rats (LHRs), which have a black hooded pattern spreading to the back, have marked hyperactive, impulsive, and inattentive behaviors [10]. The ADHD medications atomoxetine and guanfacine have been found to ameliorate LHRs’ ADHD-like behaviors [10]. Moreover, LHRs have been observed to exhibit an altered expression of several genes associated with ADHD [5,10,11]. Furthermore, compared with Wistar rats, which are much calmer than LHRs in terms of behaviors, LHRs have been reported to have a shortened axon initial segment (AIS), which is relevant in neuronal excitability in the medial prefrontal cortex (mPFC) [12]. Shortened AIS was also observed in the PFC of 6-hydroxydopamine-induced ADHD model mice [12].

Research has shown that the number of cases of NDDs, including ADHD, has been increasing worldwide in recent years [13], which is at least in part due to the increased awareness of NDDs and positive attitudes toward their diagnosis. However, changes in the social environment surrounding children could likely be responsible for the substantial increase in the incidence of ADHD; increased use of digital media [14], the prevalence of computer games [15,16,17], cell phone use [18], and decreased outdoor play have been implicated in the increase in ADHD cases. Outdoor play in green spaces has in fact been reported to reduce ADHD symptoms [19,20]. Furthermore, low fertility rates, which are common in developed countries [21], might be correlated with the increased incidence of ADHD because of decreased social stimulation from children of the same age.

Thus, in addition to genetic factors, environmental factors may be responsible for ADHD symptoms. In fact, environmental and behavioral therapies used in the treatment of ADHD [8,22,23,24] have been found to be effective to some extent. Enriched environments (EEs) have been reported to reduce emotional reactivity, ameliorate abnormal NDD-like behaviors, and enhance cognitive functioning of animals [23]. Clinically, standalone environmental and behavioral therapies may not be much effective, but when combined with pharmacological treatment, the therapies provide modest advantages for non-ADHD symptom and positive functioning outcomes [22].

To gain further insights into these matters, this study examined whether changes in the rearing environment of LHRs alter ADHD-like behaviors and gene expression, such as rearing with many conspecifics or in large cages with an EE. Rearing in an EE has been reported to change behaviors of LHRs compared to isolated barren conditions; isolation rearing enhanced activity in environmental novelty, whereas rearing in EE accelerated habituation to novelty and improved spatial learning and memory [25]. Long-Evans rats reared in an EE displayed reduced motor activities, while altering dopamine clearance rates in the mPFC [26]. In contrast, there is a report demonstrating that EE rearing of SHRs is ineffective on ADHD-like behaviors [27]. Mice reared in an EE have been reported to become less anxious and hyperactive [28]. These reports prompted us to investigate whether EEs ameliorate LHRs’ ADHD-like behaviors and whether EEs alter neuronal activities in the mPFC. The present results showed that rearing in an EE suppressed ADHD-like behaviors and decreased the expression of such immediate-early genes (IEGs) as cFos, an indicator of neuronal activity, in the prelimbic region (PrL), a subregion of the mPFC.

## 2. Materials and Methods

### 2.1. Animals

All animal experiments were conducted in accordance with the Guidelines of the Ethics Committee for Animal Experimentation of Ehime University, Matsuyama, Japan (approved reference number: 05U40-1, 05U40-16). Male and female Wistar rats and LHRs were purchased from Clea Japan (Tokyo, Japan) and Kyudo (Saga, Japan), respectively, more than 20 years ago, and they have been bred in the animal experiment center of Ehime University Medical School. Only male rats were used in this study to eliminate the influence of the estrous cycle in females. The male pups were reared with their parents and then separated from them on postnatal day 23 (PND23). The rats were housed (4 rats per standard cage measuring 40 × 25 × 14 cm [width × depth × height]) in a 12 h light/dark cycle (lights on at 7:00; lights off at 19:00) in a temperature (25 °C)-controlled animal facility. Food and water were provided ad libitum. In some experiments, LHR pups were reared in one of the following three conditions until they were 9 weeks old (Figure 1): (1) a standard environment called 4S, in which 4 rats were reared in a standard cage; (2) a standard grouped environment (12S), in which 12 rats were reared in a large flat cage (90 × 40 × 14 cm); or (3) an EE (4E or 12E), in which 4 or 12 rats were reared in a large, tall cage (95 × 55 × 115 cm) with a running wheel, four stages, three wooden pens, a wooden ladder, plastic slopes, a climbing steel tube, three trays with food pellets, and three trays filled with water.

Behavioral Experiments were Divided into the following Four Batches (Figure 1C).

Behavioral experiments were divided into the following four batches. In each batch, male littermates born from the same parents were divided into two groups as shown in Figure 1.

Batch 1: Batch 1 was set to determine whether rearing in 12E condition ameliorated ADHD-like behaviors of LHRs compared to those reared in 4S condition. Twenty or 24 LHR pups were reared with their parents and littermates until PND23 (Figure 1B). Then, they were randomly divided into two groups of 8 or 12 pups; one group of 8 or 12 pups was further divided into two or three groups and raised in 4S conditions, whereas the other group of 12 pups was collectively reared in a 12E condition. The rats were reared until PND63 under the same conditions. The batch 1 rearing was repeated three times as Batch 1a, 1b, and 1c, and the rats were subjected to behavioral tests (drop, Morris water maze [MWM], 7-day open-field [OF], elevated plus maze [EPM], and light and dark [L/D] box tests) from PND56 as described in Figure 1C.

Batch 2: Batch 2 was set to determine whether rearing in 12S condition ameliorated ADHD-like behaviors of LHRs compared to those reared in 4S condition. Twenty pups were randomly divided into two groups of 8 and 12 pups; one group of 8 pups was further divided into two groups and raised in 4S conditions, whereas the other group of 12 pups was reared in a 12S condition. The rats were subjected to drop and 7-day OF tests.

Batch 3: Batch 3 was set to determine whether rearing in 4E condition ameliorated ADHD-like behaviors of LHRs compared to those reared in 4S condition. Sixteen pups were randomly divided into two groups of 8 pups each; each group of 8 pups was divided into two groups and raised in 4S or 4E conditions. The rats were subjected to drop and 7-day OF tests.

Batch 4: Batch 4 was set to determine whether rearing in 4E/6-9W condition ameliorated ADHD-like behaviors of LHRs compared to those reared in 4S condition. In the 4E/6-9W condition, rats were reared in 4S conditions until PND42, then they were reared in 4E conditions until PND 71. Sixteen pups were randomly divided into two groups of 8 pups each; each group of 8 pups was further divided into two groups and raised in 4S or 4E/6-9W conditions. The batch 4 rats were subjected to drop and 7-day OF tests.

Rats of Batches 1 and 2 were dissected of their brains and adrenal glands, or subjected to fixation for immunohistochemical analyses, and/or peripheral blood collection for determination of plasma corticosterone.

### 2.2. Behavioral Tests

The behaviors of the rats were evaluated using the OF, EPM, L/D box, drop, and MWM tests. These were conducted and video recorded during the dark phase (19:00–22:00) with a video-tracking system (Ethovision XT 14; Noldus Info. Tech., Wageningen, The Netherlands) [29]. Only one behavioral test was carried out per day. The equipment for the behavioral tests was placed in a sound-attenuated room, and the experimenter remained outside the experimental room, except in the case of MWM test. On the day of behavioral testing, the rats reared in 12S or EE conditions were moved to new standard cages from its home cages and subjected to the tests, and they were returned to their home cages after the tests.

#### 2.2.1. OF

A 5 min OF test was conducted using a square box (100 × 100 cm) with 50-cm-high walls. It measured the following parameters for 5 min: total moved distance, mobile duration, frequency entering the center zone (or number of entries into the center zone) (60 × 60 cm), and latency to the first entry into the center zone [10,29]. The illumination during OF test was 70 lux on the floor of the field.

#### 2.2.2. EPM

A 10 min EPM test was used to evaluate anxiety-like, impulsive, and hyperactive behaviors [10]. The EPM apparatus was made of black acrylic plates. The arms were 50 cm in length and 10 cm in width and raised 50 cm above the base. Closed arms were walled in with 50-cm-high acrylic plates. The rats were placed in the crossroad when they started the test as described elsewhere [30]. The movement of rats were recorded for 10 min with the video-tracking system and the frequency entering the open-arms and the duration staying in the open-arms were measured. When the whole body of a rat entered one of the open-arms, it was judged that the rat entered the open-arms. The illumination was 45 lux in the central area, 70 in each open arm, and 12 in closed arms.

#### 2.2.3. L/D Box

A 5 min L/D box test was used to assess anxiety-like, impulsive, and hyperactive behaviors. The rats were tested in a homemade L/D box (50 × 25 × 25 cm) that was made of transparent acrylic plates and had two same-sized compartments with a small square opening (8 × 8 cm), one of which was covered with black tape. The rats were placed in the light compartment when they started the test and the frequency of the entry the light chamber and the duration of the stay there were measured [10]. The illumination was 1500 lux in the light chamber.

#### 2.2.4. Drop

A drop test was conducted to evaluate ADHD-like behaviors. Each rat was put on a darkly painted wooden round platform (15 cm in diameter) raised 40 cm above the base (see Figure 2A), and its movement was video recorded for 6 min. The experimenter was in the next room for 6 min, and the rat was freely moved on the floor after it fell. It was determined whether the rats were stayed on or dropped from the disc within 6 min by viewing the video [10]. The illumination was 800 lux on the floor.

#### 2.2.5. MWM

An MWM test was used to assess short-term spatial memory and cognitive function with a circular pool measuring 150 × 45 cm (diameter × height) that was filled with pure water [31,32]. Four spatial cues were set on the wall of the pool. The MWM test was conducted once in a day. Shortly after the rats were forced to swim for 90 s in the pool and then placed on a transparent platform to allow them to memorize the location for 15 s, they were released into the water again and allowed to search for the platform for 90 s. The number of rats that had been successfully arrived at the platform within 90 s were counted. The fan-shaped area set at a 90-degree angle centered on the platform was designated as the near-by zone, and the latency to the first entry into the near-by zone as well as total moved distance was measured. The illumination was 200 lux on the liquid surface.

### 2.3. Tissue Dissection

When tissues and blood samples were collected or when rats were perfused with a fixative, rats were placed in a clear acrylic case to observe their respiratory movement and euthanized by inhalation of a 100% concentration of carbon dioxide. After confirming respiratory arrest, the rat body weights were measured, the abdomen was opened, and the bilateral adrenal glands were dissected along with the surrounding connective tissues. The connective tissue was carefully removed, and the bilateral adrenal gland weights were measured. The weight (milligrams) of the adrenal glands was divided by the body weight (grams) of each rat [33]. Two mm thick slices were prepared from the rat forebrains, and the PrL was dissected from the most anterior slice by cutting the slice along the lines 2.5 mm from the top, 3 mm from the bottom, and 1.5 mm from the midline. The hypothalamus tissues were obtained from the fourth 2 mm-thick slice located at approximately 2 mm posterior from the bregma by dissecting the right hemisphere along the lines 4 mm from the midline and 4 mm from the bottom of the hypothalamus. cDNA was prepared from the tissues and subjected to quantitative real-time reverse transcription polymerase chain reaction (RT-PCR) or RNA sequencing (RNAseq). Some rats were transcardially perfused with 4% paraformaldehyde with 2 mM MgCl_2_, and the fixed brains were dissected for immunohistochemical (IHC) analysis.

### 2.4. Enzyme-Linked Immunosorbent Assay

Enzyme-linked immunosorbent assay (ELISA) was performed as previously described to measure corticosterone levels in circulation [34]. After confirming respiratory arrest by inhalation of 100% carbon dioxide, the apex of the heart was punctured, and blood was drawn. The blood was heparinized, and plasma corticosterone levels were determined using an ELISA kit (Arbor Assays, Ann Arbor, MI, USA).

### 2.5. Quantitative Real-Time RT-PCR

The left PrL tissues were homogenized, and complementary DNA was prepared for quantitative RT-PCR (qPCR) measurements as described previously [31]. The primer sequences for each gene are listed in Table 1. All gene specific RNA expression was presented based on the 2^−ΔΔC^_T_ method using the housekeeping (reference) gene glyceraldehyde 3-phosphate dehydrogenase (GAPDH) as an internal standard as described elsewhere [35].

### 2.6. RNA Sequencing (RNAseq)

An RNA library was prepared from PrL tissues dissected from three Wistar rats and three LHRs reared in the standard condition as well as from three LHRs reared in the EE condition using a TruSeq RNA Sample Prep Kit v2-Set A (Illumina, CA, USA) according to the manufacturer’s instructions. The tissues were subjected to RNAseq on a MiSeq NGS sequencer (Illumina) as described elsewhere [36]. Genes with significant differential expression (uncorrected *p*, <0.05) and log_2_(fold change) >|0.3875| were selected and analyzed using the free software R [37].

### 2.7. IHC Staining

Paraformaldehyde-fixed PFC tissues from 6 rats of 4S and 12E groups were frozen and thin sectioned (thickness, 10 µm). Four sections from each rat containing layers I to VI were immunofluorescently stained with antibodies to c-Fos and NeuN as described elsewhere [32]. The primary and secondary antibodies are listed in Table 2. Thirty-five micrographs of the PrL region were taken using a scanning fluorescence microscope (BZ-9000; Keyence, Osaka, Japan) with a ×40 objective lens and combined into single image. The number of cFos^+^/NeuN^+^ in the PrL region in the binarized micrographs were counted using ImageJ bundled with Java 1.8.0_172 (Wayne Rasband, National Institute of Health, Bethesda, ML, USA).

### 2.8. Statistical Analysis

Data were expressed as the mean ± standard deviation or standard error of mean. Group means were compared using two-tailed unpaired Student’s t test, χ^2^ test, or one- or two-way analysis of variance (ANOVA) with Tukey’s multiple comparison test. All analyses were performed using Prism 8 (GraphPad Software, La Jolla, CA, USA). *p* < 0.05 was considered significant for all tests. The single, double, triple, and quadruple asterisks in the graphs indicate statistical significance at *p* < 0.05, 0.01, 0.001, and 0.0001, respectively.

## 3. Results

### 3.1. EE Rearing Ameliorated ADHD-Like Behaviors

To evaluate the effects of rearing conditions on LHRs’ behaviors, various behavioral tests were carried out as described in Figure 1. First, a drop test was conducted to evaluate ADHD-like behaviors of Batch 1a rats (Figure 2A). In the control group (4S), 11 of the 12 rats moved restlessly on the disk and fell backward, whereas in the 12E group, most of the rats did not move actively, as if huddled on the disk, and only 1 of the 12 rats fell. When dropped from the disk, all the rats fell backward (Figure 2Aa), suggesting that the drop was due to their inattention and hyperactivity, rather than impulsivity. On the other hand, as shown in Figure 2Ab, the 12E rats moved very carefully, although many of them curiously looked down at the floor.

Next, an MWM test was conducted to examine short-term memory related to executive functions (Figure 2B). After the rats were allowed to swim freely for 90 s in the round pool, they were placed on a transparent platform under the waterline for 15 s to allow them to memorize its location. Shortly after, the rats were released again into the water pool and forced to swim for 90 s. Of the 12 4S rats and 12 12E rats, 6 and 7, respectively, successfully reached the platform; the reaching rates were not significantly different between the groups. The total distance traveled by the rats in the 4S and 12E groups did not differ, but the time to reach the nearby zone for the first time was shorter in the 12E group. Thus, rearing in EE did not improve short-term memory and cognitive functions of LHRs, despite that many reports have shown that rats reared in EE showed marked improvements in such functions [38,39].

12E and 4S groups of Batch 1b, eight animals from each group were randomly selected for behavioral tests, whereas the remaining four were subjected to biochemical analysis. OF tests were conducted for 7 consecutive days (Figure 2C). On the first day, total distance traveled, duration of activity, and number of center zone entries were not significantly different between the two groups. However, the latency to the first entry into the center zone significantly differed between them: it was shorter in the 12E group than in the 4S group. From the second day onward, total distance traveled, duration of movement, and number of center zone entries decreased in the 12E group compared with the 4S group. The 4S group showed a hardly habituated response and continued hyperactivity.

Then, 12 rats were reared from PND23 in a standard environment in a large flat cage (Batch 2, Figure 1C; 12S group) to determine whether rearing with many conspecifics could improve ADHD-like behaviors (Figure 3A). After PND56, the rats were subjected to a 7-day OF. The results showed that ADHD-like behaviors did not improve in a standard rearing environment even with 12 rats in a cage. On the other hand, when the animals were reared in an EE in a group of four (4E) and subjected to an OF test (Batch 3, Figure 1C), the 4E group showed decreased motor activities from the third day on (Figure 3A).

Whether ADHD-like behaviors improve when rats are reared in an EE with four rats after 6 weeks of age (4E/6–9w), which corresponds to the adolescence period, was also examined (Batch 4, Figure 1C). PND23 is considered to correspond to early childhood, but diagnosis of ADHD in such early childhood is usually difficult [40]. Therefore, rearing in EE was initiated from PND42 was carried out and the effects on rats’ behaviors were examined whether growing in EE could be a realistic therapeutic intervention. Eight rats from the 4E/6–9w and 4S groups were subjected to a drop test (Figure 3B) and OF test for 7 consecutive days (Figure 3C). In the drop test, all rats in the 4S group fell, whereas only one rat in the 4E/6–9w group did. The OF test showed that the total moved distance, mobile duration, and number of entries into the center zone decreased in the 4E/6–9w rats compared with the 4S ones.

### 3.2. EE Rearing Did Not Increase Anxiety

It has been reported that increased anxiety causes behavioral inhibition in both humans and animals [41,42]. Rearing rats in a large, tall cage with various equipment for an EE may lead to injuries due to falling from a height. This danger may have induced stress in the rats, while increasing their anxiety and suppressing their behavioral activities. To evaluate the anxiety levels of the rats reared in different conditions, EPM and L/D box tests were performed (Figure 4A,B). Furthermore, weights of adrenal glands were measured, and expression of corticotropin-releasing hormone (CRH) messenger RNA (mRNA) in the hypothalamus was examined (Figure 4C–E) [43].

The 4S, 12E, and 12S LHRs were subjected to EPM (Figure 4A) and L/D box (Figure 4B) tests to assess their anxiety-like behaviors. The duration and entry frequencies in or into the open arms did not significantly differ between the groups. In the L/D test, the 12E LHRs spent a significantly longer time in the light chamber, but the frequencies of entry into the light chamber were not different between the groups. These results suggest that the 12E rats had weakly decreased anxiety compared with the 4S and 12S groups. As shown in Figure 2C, on day 1 of the OF test, the latency to the first entry into the center zone was shorter in the 12E group than in the 4S group, which also suggests a weak decrease in anxiety in the 12E group.

Although the adrenal gland weight (milligrams)/body weight (grams) ratio of the 4S LHRs was significantly lower than that of the Wistar rats, the ratio of the 12E LHRs was increased to a level comparable with that of the Wistar rats (Figure 4C). However, the ratio increased to approximately the same extent in the 12S group as in the 12E group, suggesting that the increase in adrenal gland weight was due to stress caused by rearing with many conspecifics, but not by the EE. In addition, plasma corticosterone levels did not change between the 4S and 12E rats (Figure 4D), and CRH mRNA expression in the hypothalamus (Figure 4E) was not significantly changed between the 4S and 12E rats.

### 3.3. Effects of the Rearing Conditions on Gene Expression in the PrL

The mPFC, including the PrL, is a brain region that has been implicated in ADHD-like behaviors [12,44,45]. PrL tissues were obtained from three 4S LHRs and three 12E LHRs as well as three 8-week-old Wistar rats that were reared in the same standard cage as the 4S LHRs. mRNA was purified from the tissues and subjected to RNAseq. The RNAseq revealed the presence of the gene cluster that is characteristically highly expressed in 4S LHRs, but not in Wistar rats and 12E LHRs (noted as “Cluster 1, 4S > Wistar ≈ 12E” in Figure 5A). Volcano plot analysis was performed to clarify the specific genes differentially expressed among the three groups (Figure 5B). Cluster 1 included genes *Arc*, *Egr1*, *Egr2*, *Egr4*, *Fos*, *Fosb*, and *Junb*, which are IEGs, and 4S LHRs expressed these IEGs at higher levels than Wistar rats and 12E LHRs (Figure 5C). No significant differences between the Wistar rats and 12E LHRs in terms of the IEG expression were observed. Cluster 2 included genes such as *ADPRHL1* (*ADP-Ribosylhydrolase Like 1*), *GPR22* (*G Protein-Coupled Receptor 22*), and *GIMAP6* (*GTPase, IMAP Family Member 6*) that Wistar rats and 12E LHRs expressed at higher levels than 4S LHRs. Detailed RNAseq results are listed in Appendix A.

### 3.4. Comparison of IEG Expression in the PrL among the Three LHR Groups

Tissues were collected from the PrL of the 4S, 12E, and 12S LHRs for qPCR to see whether the expression of IEGs in the PrL is correlated with ADHD-like behaviors (Figure 6A). In accordance with the RNAseq results, *Fos*, *Arc*, and *Egr2* mRNA expression levels were all suppressed in the 12E rats. 12S rats expressed these IEGs at higher levels than 12E ones. IHC staining of cFos and the neuronal marker NeuN revealed a decrease in the number of cFos^+^ neurons in the PrL of the 12E rats compared with the 4S rats (Figure 6B,C). 

## 4. Discussion

In this study, we employed LHRs as ADHD model animals and investigated the effects of rearing environments on their behaviors. LHRs are more hyperactive, inattentive, and impulsive than SHRs, as revealed by various behavioral test batteries [10]. Twelve LHRs were reared in an EE that consists of a large, tall cage with various playthings. The results showed that rearing in the EE led to an improvement in their ADHD-like behaviors. On the other hand, rearing with many conspecifics was not an essential factor for the amelioration of ADHD-like behaviors, suggesting that non-social enrichment more effectively improved ADHD-like behaviors than social enrichment. In addition, ADHD-like behaviors were suppressed not only by rearing in the EE from PND23 but also by rearing in the EE from PND42, an adolescence period. RNAseq, qPCR, and IHC staining revealed that expression of IEGs, such as cFos, in the PrL located in the mPFC was decreased when ADHD-like behaviors were ameliorated by rearing LHRs in the EE condition.

Outdoor play and numerous behavioral interventions have been applied to treat ADHD cases [19,20,23]. Although clinical trials have shown that stand-alone behavioral therapy is less effective than stand-alone pharmacological therapy, it could exert synergistic ameliorative effects when administered with medications [46]. Once the underlying molecular and cellular mechanisms of the ameliorative effects of nonpharmacological therapies have been clarified, understanding the synergistic effects would be beneficial. Research has shown that SHRs, a prevalent ADHD rat model, reared in EE conditions demonstrate accelerated habituation in OF tests and amelioration in spatial cognitive functions in MWM tests [47]. Another study found that rearing in an EE ameliorated the hyperactivity and inattention, but not impulsivity, of SHRs [48]. Nonetheless, the ineffectiveness of EE rearing on ADHD-like behaviors has also been reported [27]. This study is, however, supported the notion that EE decreased ADHD-like behaviors of rats.

In the drop test, the 4S LHRs repeatedly behaved restlessly on a 15 cm-diameter disk placed at a 40 cm height, and most of the rats fell backward within 6 min. However, most 12E LHRs showed fewer movements while placing four limbs stably on the disc for 6 min. A cliff avoidance reaction test, which partially resembles the drop test, has often been employed to evaluate ADHD-like behaviors, especially impulsivity, in animal models [7,49]. In a cliff avoidance reaction test, ADHD model animals fall in a forward motion, and such falling has been interpreted as an impulsive behavior. On the other hand, in the drop test used in this study, the rats fell backward, which may have likely been caused by the inattention, rather than impulsivity, of the rats. mPFC neurons have been implicated in the suppression of hyperactive and impulsive behaviors; however, based on the drop test results obtained in this study, they may also be responsible for the inhibition of inattentive behaviors as has been described [50].

In the OF test, the 4S LHRs were not habituated to the OF arena and remained hyperactive throughout the 7 consecutive test days. By contrast, the 12E LHRs were soon habituated and became less active beginning on the second day; this changed behavioral attitude is almost the same as those previously observed in Wistar rats [10]. The MWM test was conducted to evaluate spatial cognitive functions and very short-term memory, which are the executive functions of the PFC [45]. In ADHD cases, such executive functions of the PFC are frequently impaired, although the impairment is not a typical symptom of ADHD [51]. The MWM test results showed that EE rearing apparently did not affect the short-term memory of the LHRs, despite many reports describing improved memory, learning, and cognitive functions in animals reared in EE conditions by increased expression of brain-derived neurotrophic factor [38], enhanced neurogenesis [39], and so on. Chronic stress can cause cognitive decline [52,53,54] in rats and humans. LHRs reared in 12E condition have been chronically stressed as indicated by enlarged adrenal glands, which might have inhibited their improvement of cognitive functions.

Although the 12E LHRs were calmer in behaviors than the 4S LHRs, they did not restrain their mobile activities showing the comparable mobile activities of the 4S LHRs on the first day of the OF test. The latency to the first entry into the center zone of the 12E rats was shorter than that of the 4S rats in the OF test on the same day. Because the decreased activity of the 12E rats could be attributed to their increased anxiety [41,42], EPM and L/D box tests were conducted to assess the anxiety levels of the rats. Both tests showed a weak decrease in the anxiety levels of the 12E rats, indicating that the decreased activities in the OF test are unlikely caused by increased anxiety among the rats.

RNAseq revealed the absence of broad differences in gene expression in the PrL in the mPFC between the Wistar rats reared in a standard condition and the 4S LHRs. Among the small number of differentially expressed genes, IEGs, including *Fos*, *Egr2*, and *Arc*, which are responsible for increased neuronal excitability and plasticity [55], were expressed by the 4S LHRs at higher levels compared with the Wistar rats. Such increased IEG expression in the 4S LHRs was almost abolished in the 12E LHRs, whose expression was reduced to levels that were almost comparable with those in the Wistar rats. In fact, the finding that the behavioral characteristics of the 12E LHRs are very similar to those of the Wistar rats in the drop test and the 7-day OF test has been described elsewhere [10]. These observations suggest that increased expression of IEGs is correlated with ADHD-like behaviors. Indeed, research has shown that increased cFos expression in the mPFC of mice leads to hyperactivity and impulsive behaviors [56].

The mPFC has been implicated in executive functions and behavioral inhibition, and mPFC dysfunction may be correlated with ADHD-like behaviors [44,56]. Abnormal molecular structures of the AIS in the mPFC of LHRs have been previously reported [12], likely a suggestive of disturbed mPFC functions of LHRs. The increase in the number of cFos^+^ neurons has specifically been observed in the mPFC of ADHD model mice prepared by repeated inhalation of an anesthetic (sevoflurane) during the developmental stage, and chemogenetical inhibition of the mPFC excitatory neurons of the sevoflurane-induced ADHD model partially rescued the deficit [56]. Conversely, chemogenetical activation of excitatory neurons in the mPFC caused impulsive behaviors in the ADHD model mice. These reports suggest that increased activities of the mPFC accompanying elevated IEG expression are correlated with ADHD-like behaviors. EEs have been found to increase or enhance the development of the inhibitory γ-aminobutyric acid (GABA) system in the visual cortex of normal rat pups [23]. Although whether GABAergic neuron activities in the mPFC are enhanced by rearing in EEs remains unclear, they are a potential mechanism underlying the EE-induced decrease in the number of cFos^+^ neurons.

The 4S LHRs had a smaller adrenal gland weight/body weight ratio compared with the Wistar rats. The ratio in the 12E and 12S LHRs was increased to the level observed in the Wistar rats. This outcome indicates that rearing with many rats may induce stress in the immature rats. The stressful rearing, enlarged adrenal glands, and ADHD-like behaviors may not be significantly correlated. Although decreased salivary cortisol levels and attenuated cortisol-secreting responses have been reported in ADHD cases [57,58], these are not ADHD-specific changes and instead are related to oppositional defiant disorder and conduct disorder, which are often associated with ADHD cases [59,60]. Chronic stress alters neuronal activities in the mPFC in both humans and rodents [52]. Chronic social defeat stress induced elevated cFos expression in the mPFC of adolescent mice [61]. Acutely administered corticosterone increases expression of *Fos* and *Arc* in the mPFC [62]. Thus, many reports have demonstrated the intimate correlation among stress, glucocorticoids, and increased mPFC activities. However, the present results suggest that the expression of IEGs or the number of cFos^+^ neurons in the PrL in the mPFC are not directly correlated with experienced stress or the size of adrenal glands; 4S rats with the small adrenal glands and the 12S rats with the large ones expressed IEGs at higher levels than the 12E rats with the large ones. Thus, mechanisms underlying the changes in IEG expression in LHRs reared in EE conditions are at present still to be elucidated.

## 5. Conclusions

This study suggests that nurturing children in EEs will improve the behavioral problems of ADHD—at least partly by normalizing the activities of the PrL. Furthermore, nurturing in EEs from the adolescence period may still be an effective intervention to facilitate said improvement. Rearing LHRs with many conspecifics did not significantly ameliorate ADHD-like behaviors, suggesting that stimulation from the environment is more effective in improving ADHD-like behavior than stimulation from interaction with many children of the same age. Although medications are the first-line treatment for ADHD [63], the most prevalent psychostimulant methylphenidate may have negative impacts physically and mentally [64,65,66]. With such risks of pharmacological interventions, it may be necessary to become more proactive in nurturing ADHD children in an EE.

## Figures and Tables

**Figure 1 cells-11-03649-f001:**
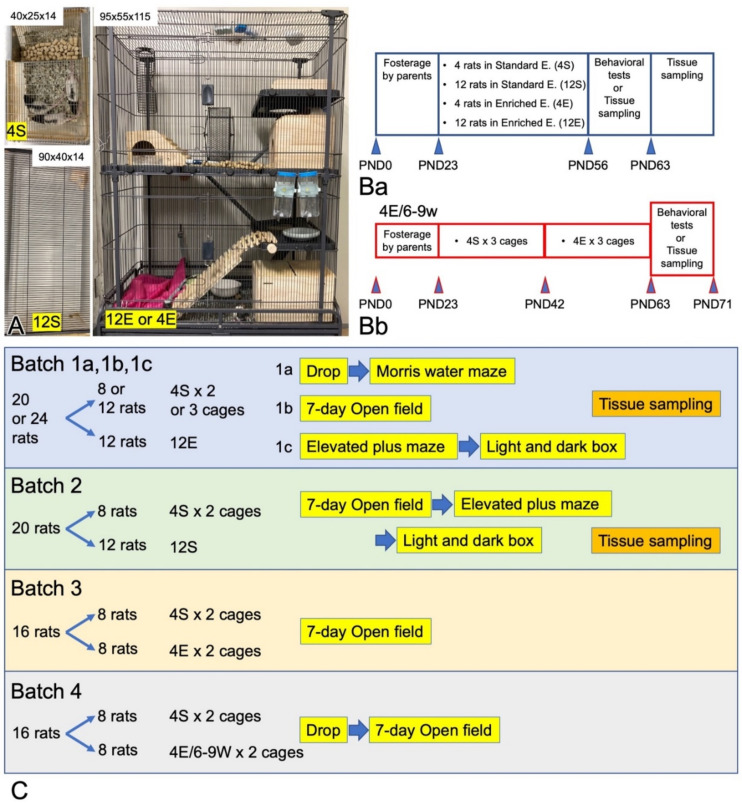
Rearing cages for LHRs and experimental protocols. (**A**) Rearing conditions. LHRs were reared in one of four conditions, namely, 4S, 12S, 12E, and 4E. Under 4S, 4 LHRs were kept in a standard cage measuring 40 × 25 × 14 cm [width × depth × height]; 12S, 12 LHRs were kept in a large flat cage (90 × 40 ×14 cm); and 12E or 4E, 12 or 4 LHRs were kept in a large, tall cage (95 × 55 × 115 cm) with four stages, a running wheel, three wooden pens, and other features. (**B**) Rearing schedule. LHR pups were reared with their parents until PND23 and then kept in each rearing condition up to PND63 (**Ba**). (**Bb**) The pups of 4E/6-9w group were kept in a standard cage until PND42 and then reared in a large, tall cage until PND71. During the behavioral test days, the rats were kept in the same cages. (**C**) Four kinds of Batches of rats were prepared, reared, and subjected to behavioral tests. Two groups were formed in each batch by bisecting male littermates born from the same parents.

**Figure 2 cells-11-03649-f002:**
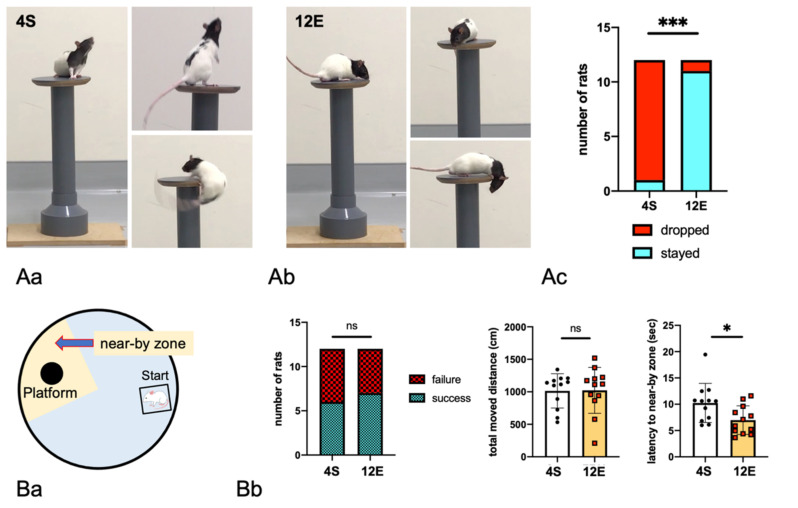
Behaviors of LHRs from Batch 1 reared in the 12E and 4S conditions. (**A**) Representative video clip images of the behaviors of 4S and 12E LHRs during a drop test. (**Aa**) The 4S LHRs showed frequent movement on a disk placed at a height of 40 cm and fell from the disk. Images of a 4S rat that slipped and fell backward are shown. (**Ab**) The 12E LHRs were much calmer than the 4S ones, and they did not fall even though they were curious about the space below. (**Ac**) Statistical findings on the drop test. A total of 11 of the 12 4S rats and 1 of the 12 12E rats fell. χ^2^ test and Fisher’s exact test. (**B**) An MWM test was conducted to evaluate the very short-term memory of the rats. (**Ba**) Settings of the arena. After the rats were trained to memorize the location of the transparent platform under the waterline, they were subjected to the MWM test for 90 s. Panel (**Bb**) shows the number of rats that successfully reached the platform within 90 s, their total distance moved, and their latency to the first entry into the nearby zone. Unpaired two-tailed t test, *n* = 12. (**C**) Batch 1b rats were subjected to an OF test conducted for 7 consecutive days. (**Ca**) Representative heatmaps displaying movement of one rat of 4S or 12E groups on days 1 and 7. All other heatmap data are shown in Appendix A. (**Cb**) On the first day, significant differences in the OF test parameters between 4S and 12E rats were not observed, except for the latency to the first entry into the center zone. In the subsequent days, significant differences between the two groups, indicating that the 4S rats were more hyperactive than the 12E rats, were noted. Two-way analysis of variance (ANOVA) with Tukey’s multiple comparison test, *n* = 8. The single, double, triple, and quadruple asterisks in the graphs indicate statistical significance at *p* < 0.05, 0.01, 0.001, and 0.0001.

**Figure 3 cells-11-03649-f003:**
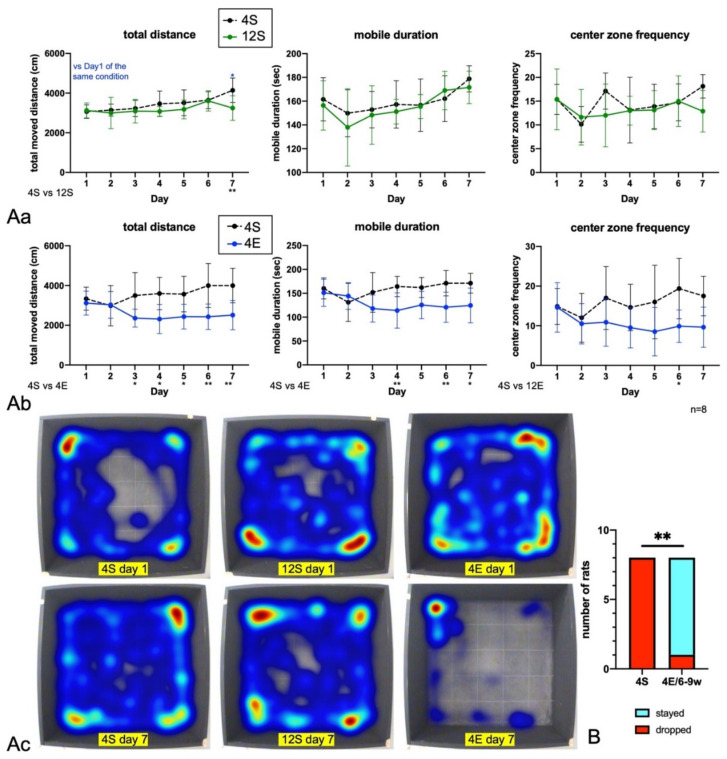
Day OF test data of LHRs of Batches 2, 3, and 4 (*n* = 8). (**A**) (**Aa**) The behaviors of the 4S and 12S LHRs were almost the same throughout the 7-day OF test. (**Ab**) The 4E rats were less active than the 4S ones. (**Ac**) Representative heatmaps of the OF test for each group. On day 7, only the 4E rats exhibited decreased activity. Two-way ANOVA with Tukey’s multiple comparison test. (**B**) The drop test showed that the 4E/6–9w LHRs were more attentive than the 4S control rats. χ^2^ test and Fisher’s exact test. (**C**) The 7-day OF test showed that the 4E/6–9w LHRs had a less active nature compared with the 4S rats. The results of tests were analyzed with two-way ANOVA with Tukey’s multiple comparison test. The single, double, triple, and quadruple asterisks in the graphs indicate statistical significance at *p* < 0.05, 0.01, 0.001, and 0.0001.

**Figure 4 cells-11-03649-f004:**
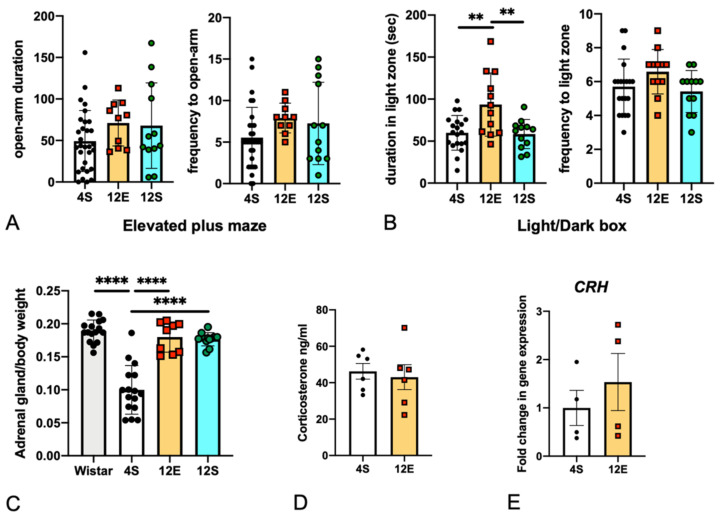
Behavioral test data (EPM and L/D box tests) of the 4S, 12E, and 12S rats of Batches 1 and 2, and weights of the adrenal glands, plasma corticosterone levels, CRH mRNA expression in the hypothalamus. (**A**) The EPM test did not produce significant results for the 4S (*n* = 30), 12E (*n* = 10), and 12S (*n* = 12) groups. One-way ANOVA with Tukey’s multiple comparison test. (**B**) L/D box test. The 12E rats (*n* = 12) remained longer in the light chamber than did the 4S (*n* = 20) and 12S (*n* = 12) rats. One-way ANOVA with Tukey’s multiple comparison test. (**C**) The adrenal gland weight (milligrams)/body weight (grams) ratio of the 4S LHRs (*n* = 24) was lower than the ratios of the other groups. Wistar rats (*n* = 10) reared in the 4S condition had larger adrenal glands compared with the 4S LHRs. The 12E (*n* = 9) and 12S (*n* = 12) LHRs had almost the same weight of adrenal glands. ANOVA with Tukey’s multiple comparison test. (**D**) The plasma corticosterone levels of the 4S and 12E LHRs (*n* = 6) were almost the same. Unpaired two-tailed t test. (**E**) The CRH mRNA level in the hypothalamus of 12E LHRs and the 4S ones. One-way ANOVA with Tukey’s multiple comparison test, *n* = 4. The double and quadruple asterisks in the graphs indicate statistical significance at *p* < 0.01 and 0.0001.

**Figure 5 cells-11-03649-f005:**
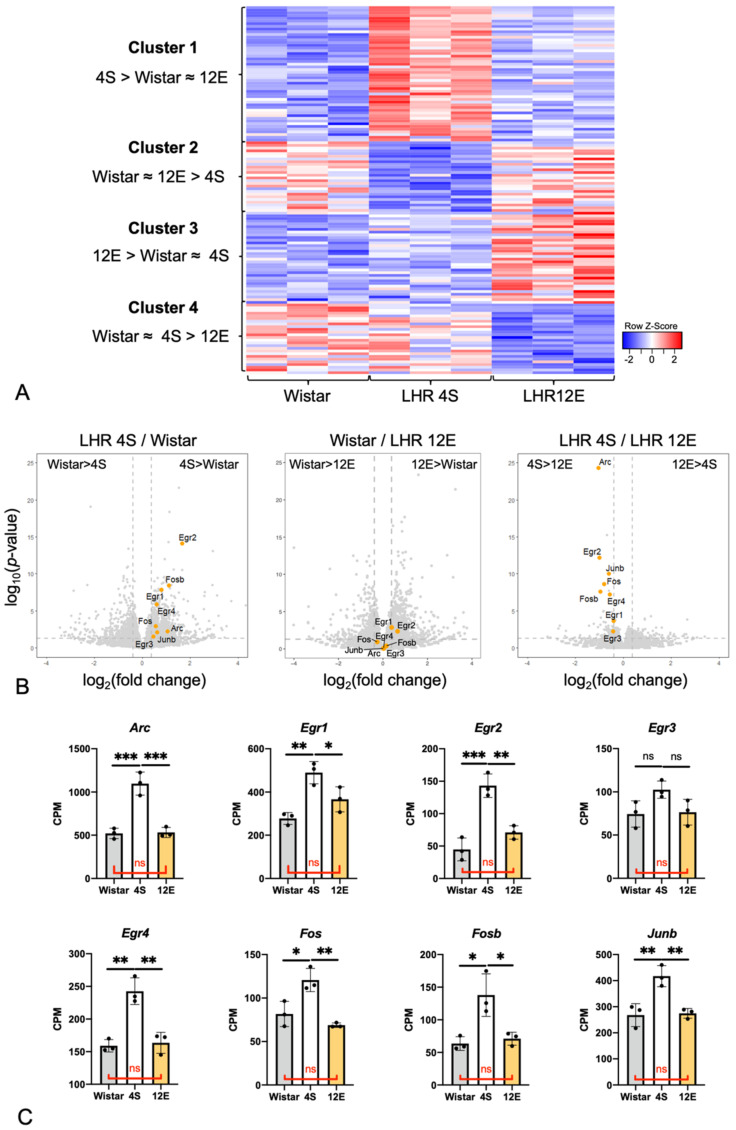
Gene expression in the PrL of Wistar rats, 4S LHRs, and 12E LHRs (*n* = 3, each) was investigated with RNAseq. (**A**) Hierarchical clustering of gene expression of the three groups. Cluster 1 is the one expressed more strongly by the 4S LHRs relative to the other two groups. (**B**) Volcano plot analysis showed stronger expression of IEGs, including *Fos*, *Egr2*, and *Arc*, in the 4S and 12E LHRs compared with the Wistar rats. The 12E LHRs and Wistar rats demonstrated nearly the same levels of IEG expression. (**C**) 4S LHRs expressed *Arc*, *Egr1*, *Egr2*, *Egr4*, *Fos*, *Fosb*, and *Junb*, at higher levels than Wistar rats and 12E LHRs. One-way ANOVA with Tukey’s multiple comparison test. Genes belonging to each cluster are listed in Appendix A. The single, double, and triple asterisks in the graphs indicate statistical significance at *p* < 0.05, 0.01, and 0.001.

**Figure 6 cells-11-03649-f006:**
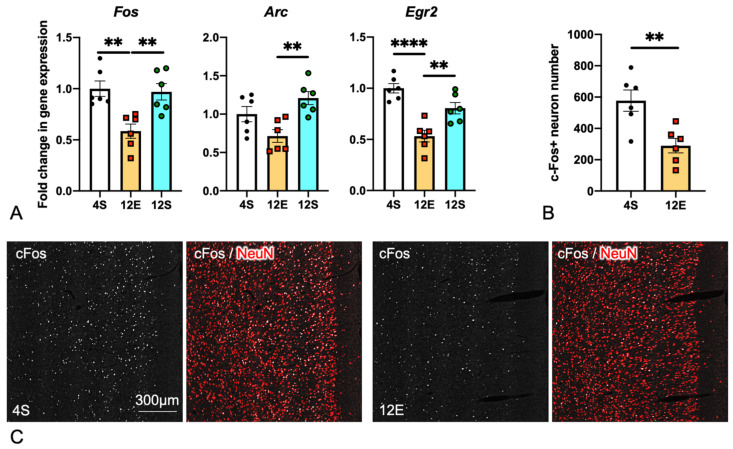
Expression of *Fos*, *Arc*, and *Egr2* in the mPFC of 4S (*n* = 12), 12E (*n* = 6), and 12S (*n* = 6) LHRs. (**A**) qPCR revealed that the 12E rats expressed mRNA encoding *Fos*, *Arc*, and *Egr2* at weaker levels compared with the 4S and 12S rats. One-way ANOVA with Tukey’s multiple comparison test. (**B**) The number of cFos^+^/NeuN^+^ neurons in the mPFC was counted on immunohistochemically stained specimens of the 4S and 12E rats. Unpaired two-tailed t test. (**C**) Representative micrographs of the mPFC of the 4S and 12E groups stained with antibodies to cFos and NeuN. Enlarged micrographs are shown in Appendix A. The double and quadruple asterisks in the graphs indicate statistical significance at *p* < 0.01 and 0.0001.

**Table 1 cells-11-03649-t001:** Primer sequences.

qPCR Primers	Fwd	Rvs
Arc (Activity Regulated Cytoskeleton Associated Protein)	GGCATCTGTTGACCGAAGTGT	CACATAGCCGTCCAAGTTGTTCT
Fos (Fos Proto-Oncogene)	AGCCGACTCCTTCTCCAGCA	AAGTTGGCACTAGAGACGGACAGAT
CRH (Corticotropin Releasing Hormone)	AGGGAAGTCTTGGAAATG	CCGATAATCTCCATCAGTT
Egr2 (Early Growth Response 2)	GATCTGCATGCGAAACTTCAG	GCAAACTTACGGCCACAATAG
GAPDH	GAGACAGCCGCATCTTCTTG	TGACTGTGCCGTTGAACTTG

**Table 2 cells-11-03649-t002:** Antibodies.

Antigen	Antibody	Concentration	Source
cFos	rabbit monoclonal (9F6)	1/500	Cell signaling, Danvers, MA
NeuN	guinea pig polyclonal	1/500	Merck millipore, Darmstadt, Germany
rabbit IgG	donkey polyclonal; labelled with DyLightL 488	1/500	Jackson ImmunoResearch Labs, West Grove, PA
guinea pig IgG	donkey polyclonal; labelled with Cy5	1/500	Jackson ImmunoResearch

## Data Availability

The data shown in this study are available from the corresponding author on reasonable request.

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
