# Peer review of "Rearing in an Enriched Environment Ameliorates the ADHD-like Behaviors of Lister Hooded Rats While Suppressing Neuronal Activities in the Medial Prefrontal Cortex"

_cells, 2022, doi:10.3390/cells11223649_

Round 1

Reviewer 1 Report

In this study, the authors have performed behavior and genetic analyses using Lister hooded rats (LHRs) as ADHD model animals, and suggested that growing in an enriched environment (EE) from childhood or adolescence improves ADHD-like behaviors via the suppression of neuronal activity in the mPFC. Overall, albeit the obtained results are modest, their analytic approaches are scientifically sound. The topic is timely and covers recent findings comprehensively. The manuscript is well written. Tables and figures are informative. This manuscript will be valuable in this research field.

Author Response

Response to Reviewer 1

In this study, the authors have performed behavior and genetic analyses using Lister hooded rats (LHRs) as ADHD model animals, and suggested that growing in an enriched environment (EE) from childhood or adolescence improves ADHD-like behaviors via the suppression of neuronal activity in the mPFC. Overall, albeit the obtained results are modest, their analytic approaches are scientifically sound. The topic is timely and covers recent findings comprehensively. The manuscript is well written. Tables and figures are informative. This manuscript will be valuable in this research field.

[First of all, we would like to express our deepest gratitude to Reviewer for the supporting comments and suggestions despite busy days. We will further develop this research.]

Reviewer 2 Report

In this work, Utsonomiya et al investigated how environmental modifications in the post-natal life affected the behavior of Lister hooded rats, an interesting rodent model for ADHD. This is a highly original piece of data on a very relevant topic. It provides interesting behavioural findings that complement and extend the existing literature. The main conclusions of the study (differences between social enrichment vs physical enrichment) open new perspectives for future research. 

The presentation of the objectives and the experimental approach is concise and clear. The results section is written in a very logical manner. The discussion is also easy to read but too long in my opinion. 

Here they are some minor comments for the authors to improve their work

1. Results for anxiety: although I do understand the rationale, these data are a little misleading. I would suggest to simplify them and remove the adrenal gland weight and POMC mRNA. I would present only the corticosterone levels as well as mRNA for CRH after the behavioral data (EPM-L/D box)

2. Authors focus their gene expression analysis on those genes (Cluster 1) whose expression is normalised in the 12E group (and similar to wistar rats). However, one would raise the question about what are the genes in cluster 2 (those upregulated in 12E compared to 4S and similar to wistar). This data should be described somewhere 

3. IEG activation: if possible, I would like to see the layer distribution of c-fos + cells in the mPFC. In addition, authors should state in which subregion of the mPFC they perform their counting (infralimbic, prelimbic ...)

4. I would shorten the discussion on stress and focus more on the differences between social and non-social enrichment try to correlate these findings with the sparse data existing in the literature

Author Response

Response to Reviewer 2

In this work, Utsonomiya et al investigated how environmental modifications in the post-natal life affected the behavior of Lister hooded rats, an interesting rodent model for ADHD. This is a highly original piece of data on a very relevant topic. It provides interesting behavioural findings that complement and extend the existing literature. The main conclusions of the study (differences between social enrichment vs physical enrichment) open new perspectives for future research. 

[We would like to express our deepest gratitude to Reviewer for the supporting comments and suggestions despite busy days. We will continue our efforts to develop this research. Our responses are written in red and in the parenthesis.]

The presentation of the objectives and the experimental approach is concise and clear. The results section is written in a very logical manner. The discussion is also easy to read but too long in my opinion. 

[We have shortened Discussion to some extent.]
Here they are some minor comments for the authors to improve their work

  1. Results for anxiety: although I do understand the rationale, these data are a little misleading. I would suggest to simplify them and remove the adrenal gland weight and POMC mRNA. I would present only the corticosterone levels as well as mRNA for CRH after the behavioral data (EPM-L/D box)

[In response to this comment, we have deleted the data on POMC mRNA expression in the pituitary. However, despite the important and kind suggestion, we have decided not to delete the data on adrenal glands. The data are reliable ones suggesting that rearing with many conspecifics causes stress. Stress has been shown to worsen the children with developmental disorders. Thus, we wish to have the data on adrenal gland weights to discuss the effects of rearing in EE. Furthermore, according to the suggestion, we have reorganized by placing the behavioral data before the data regarding HPA axis.]

  1. Authors focus their gene expression analysis on those genes (Cluster 1) whose expression is normalised in the 12E group (and similar to wistar rats). However, one would raise the question about what are the genes in cluster 2 (those upregulated in 12E compared to 4S and similar to wistar). This data should be described somewhere 

[Thank you for the critical suggestion. We have added the description in Results to show what kinds of genes are differentially expressed in the cluster 2. More detailed data including those on clusters 3 and 4 are shown in the supplementary table 1 (an excel file).]

  1. IEG activation: if possible, I would like to see the layer distribution of c-fos + cells in the mPFC. In addition, authors should state in which subregion of the mPFC they perform their counting (infralimbic, prelimbic ...)

[We tried to count the number of cFos+ neurons in different layers as shown in the below. However, currently we do not have available antibodies to identify layers in the PFC. It takes about three weeks to purchase those antibodies. We are sorry not to respond well to this suggestion. We investigated prelimbic regions in the mPFC. We have added description regarding the subregion of mPFC.]

  1. I would shorten the discussion on stress and focus more on the differences between social and non-social enrichment try to correlate these findings with the sparse data existing in the literature

[We have deleted some sentences on stress and added some sentences on social and non-social enrichment in response to this suggestion. The two paragraphs describing on stress were combined. Overall, length of Discussion was shortened to some extent.]

Reviewer 3 Report

This manuscript addresses an interesting phenomenon of the effects of enriched environment on the ADHD-like phenotype of Lister hooded rats. The paper needs major and minor improvements as some data presented do not reflect graphs, in some cases animal number is inconsistent within the same group, quality of IHC figures does not allow to analyze them, PCR calculation should not be conducted the way it was presented and other  changes are necessary for this manuscript. I suggest to focus on these 15 concerns:

1. Can you justify postnatal day 23 to assemble animals into four rearing conditions? 2. Having open field test for 7 consecutive days diminishes the concept of the open field test. Can you provide more details why this experimental plan was chosen? 3. It is not clear how the Morris water maze was conducted. Was it filled with pure water or you added a dye to hide the underwater platform? Have you changed the location of the platform to exclude a place preference factor? How many spacial cues have you used? 4. Can you provide details how you collected the blood samples? 5. It is not clear how you collected adrenal glands. 6. You mention that you have divided the weight of adrenal glands by the bodyweight of rats and made a point it was milligrams and grams respectively. Can you justify that? 7. You stated that the blood samples you have gathered at 19:00 which is the time when rats move from day to night cycle. What was the rationale for that and how many days and animals you’ve used to complete sample collection? 8. The standard thickness of an IHC sample for cFOS stating is 50 mkm, while you’ve chosen 10 mkm. Can you justify that thickness? 9. Fig2 should be re-analysed, it is not clear what is the centre zone frequency and what does different colours on the heat map mean. If we assume that the center zone frequency is the number of entries, the heat maps on the Ca figure does not correspond to the graphs on the Cb. For example, 12E day 1 clearly looks more bright and have red insertions in the center than 4S day 7, while on the graph Cb it is showing the opposite. 10. Can you add number of animals tested in each test to your graphs? When you used dot-plot graphs surprisingly number os animals tested in different behavioural tests or biochemical assays varies within the same group. Can you please justify why have you tested different sample sizes or exclude some animals from the statistical analysis? 11. In the open field test it is unclear whether the total arena was illuminated or there was a light gradation center vs periphery. 12. Can you express mRNA level in a standardized way through ddCt (2-∆∆CT algorithm)? It is unusual to display gene expression changes as a percent to a housekeeping gene because to calculate the relative expression you have already used the housekeeping gene Ct in your calculations. I assume the data presented on graphs Ca and Cb of the Fig4 cannot be presented the way the are. 13. Can you provide more details on how you dissected hypothalamus and pituitary gland for the PCR? 14. Fig5 should be explained and labeled, it is unclear what Fig 5A outlines and how these data were gotten. This figure does not allow to examine data so this part of work is assumed missing. The concept of this figure leads to an assumption that two-way ANOVA should be conducted as a statistical test, while the authors use the one-way ANOVA. Statistical methods should be double checked too. 15. Fig6C represents an IHC analysis and 12E pictures have clearly visible torn areas that alone makes those pictures to be unsuitable for further analysis. These pictures should be replaced by good quality sections and counting should be re-done.

Author Response

Response to Reviewer 3

This manuscript addresses an interesting phenomenon of the effects of enriched environment on the ADHD-like phenotype of Lister hooded rats. The paper needs major and minor improvements as some data presented do not reflect graphs, in some cases animal number is inconsistent within the same group, quality of IHC figures does not allow to analyze them, PCR calculation should not be conducted the way it was presented and other  changes are necessary for this manuscript. I suggest to focus on these 15 concerns: 

[We would like to express our deepest gratitude to Reviewer for the constructive comments and suggestions despite your busy days. Our responses are written in red and in the parenthesis.]

[We have been instructed to submit a revised version within 5 days, and we are very sorry for that we could not fully answer to some of your comments and suggestions, while we have done our best.]

[In order to meet the deadline, we did not have time to order a professional company to correct and edit the English language. We will be sure to do proofread the English text next time.]

  1. Can you justify postnatal day 23 to assemble animals into four rearing conditions?

[LHR pups may be able to be divided from their parents on postnatal day (PND) 21, and they can drink and eat by themselves. However, their bodies seemed to be too small to seek food and water in the large and tall cages (EE condition). These are just the observations by the experimenters and there are no firm data showing that the PND23 is the best day to be subjected to each rearing conditions.]

  1. Having open field test for 7 consecutive days diminishes the concept of the open field test. Can you provide more details why this experimental plan was chosen?

[Children with ADHD are characterized by a lack of decreased activity in familiar settings (e.g., school classrooms), which is a hyperactivity symptom of ADHD. To show the amelioration in the hyperactivity by rearing rats in enriched environments, we have done the 7-day OF test. This test was also done in our previous paper (Jogamoto, Utsunomiya et al, https://doi.org/10.1016/j.neuint.2020.104857)]

  1. It is not clear how the Morris water maze was conducted. Was it filled with pure water or you added a dye to hide the underwater platform? Have you changed the location of the platform to exclude a place preference factor? How many spacial cues have you used?

[We have used pure water and the place of the platform was not changed. Four spacial cues were set on the wall of the pool. These descriptions were added to materials and methods section. We performed the MWM test not to measure spatial cognitive functions or learning or memory function, but to examine executive function (including very short-term memory of the frontal cortex.]

  1. Can you provide details how you collected the blood samples?

[We have added description regarding the blood sample collection in Materials and methods; After confirming respiratory arrest by inhalation of 100% carbon dioxide, the apex of the heart was punctured, and blood was drawn.]

  1. It is not clear how you collected adrenal glands.

[We have added description regarding the collection of adrenal glands; After confirming respiratory arrest, the abdomen was opened, and the bilateral adrenal glands were removed along with the surrounding connective tissues. The connective tissue was carefully removed, and the bilateral adrenal gland weights were measured.]

  1. You mention that you have divided the weight of adrenal glands by the bodyweight of rats and made a point it was milligrams and grams respectively. Can you justify that?

[It might be more rational to use grams for both weights of adrenal glands and rat bodies. However, in that case, description “000 (three zeros)” must be added to the Y-axis of the graph. It may be a little hard to see the graphs. Therefore, we have chosen the milligrams for adrenal gland weights.]

  1. You stated that the blood samples you have gathered at 19:00 which is the time when rats move from day to night cycle. What was the rationale for that and how many days and animals you’ve used to complete sample collection?

[All the behavioral tests were started from 19:00, and we wished to know the biological backgrounds that were supposed to affect the changes in behaviors. Therefore, collection of not only blood samples but also adrenal gland, and brains were started at 19:00. Collection of samples was done in one day in each Batch. The numbers of animals were described in figure legends.]

  1. The standard thickness of an IHC sample for cFOS stating is 50 mkm, while you’ve chosen 10 mkm. Can you justify that thickness?

[Despite the kind and detailed comment, we are afraid to think that 50-mkm thickness may be too thick for antibodies to infiltrate into the total tissues, and consequently there should be unstained or unevenly stained regions. In our experience, 10-mkm thick specimens are always evenly immunostained when the specimens are stuck to slide glasses. 20-mkm thick sections can be evenly immunostained when a floating method is employed. However, 20-mkm sections may be too thick to observe evenly though fluorescence microscopy with x 40 lens.]

  1. Fig2 should be re-analysed, it is not clear what is the centre zone frequency and what does different colours on the heat map mean. If we assume that the center zone frequency is the number of entries, the heat maps on the Ca figure does not correspond to the graphs on the Cb. For example, 12E day 1 clearly looks more bright and have red insertions in the center than 4S day 7, while on the graph Cb it is showing the opposite.

[We have added description what the frequency means (number of entries) in the Materials and methods section. Color key showing what colors mean was added to Figure 2. The figure shown in Ca was replaced with more suitable one that corresponds to the average. We also showed all the heat maps of 24 rats used for Figure 2 data in supplementary Figure 1.]

  1. Can you add number of animals tested in each test to your graphs? When you used dot-plot graphs surprisingly number os animals tested in different behavioural tests or biochemical assays varies within the same group. Can you please justify why have you tested different sample sizes or exclude some animals from the statistical analysis?

[The number of animals could not be standardized for the following reasons. The main reason for the different number of experiments is that the number of rats is limited by the fact that littermates from the same parents were divided into two equal groups to create a standard rearing group and an enriched environment rearing group. This was aimed to avoid the effects of different manner in care by their parents. [For example, Liu et al. said “variations in maternal care affect the development of individual differences in neuroendocrine responses to stress in rats” (doi: 10.1126/science.277.5332.1659).] Since this point was not adequately described, an explanation was added to the Materials and methods section. The second reason is the failure of the experiments. These include failure to dissect the adrenal glands or mPFC (destruction of the tissues). In addition, three rats in the enriched environment rearing group were excluded from behavioral tests and tissue dissection due to leg injuries from falling or fighting. Because of concerns that prolonged behavioral experiments might adversely affect the circadian rhythms of rats, only randomly selected 8 rats from each 12 rat group were subjected to open-field test so that the test are to be completed between 7:00 PM and 10:00 PM. We have described the number of animals in each legend.]

  1. In the open field test it is unclear whether the total arena was illuminated or there was a light gradation center vs periphery.

[Although the illuminance of Arena is measured at the center, the lighting was from directly above, and we do not believe that there is any particular difference in illuminance between the ambient and the center. No obvious shadow areas were visible in the open-field arena (please see Fig. 3Ac, for example).]

  1. Can you express mRNA level in a standardized way through ddCt (2-∆∆CT algorithm)? It is unusual to display gene expression changes as a percent to a housekeeping gene because to calculate the relative expression you have already used the housekeeping gene Ct in your calculations. I assume the data presented on graphs Ca and Cb of the Fig4 cannot be presented the way the are.

[In response to this request, we have replaced all the qPCR data graphs with those expressed in the ddCt basis.]

  1. Can you provide more details on how you dissected hypothalamus and pituitary gland for the PCR?

[We added detailed description regarding the dissected positions of the hypothalamus in Materials and methods section. Another reviewer requested us to remove the data on POMC, we did not describe the method to dissect the pituitary.]

  1. Fig5 should be explained and labeled, it is unclear what Fig 5A outlines and how these data were gotten. This figure does not allow to examine data so this part of work is assumed missing. The concept of this figure leads to an assumption that two-way ANOVA should be conducted as a statistical test, while the authors use the one-way ANOVA. Statistical methods should be double checked too.

[Some descriptions were added to Materials and methods. The complete data used for Fig. 5A are newly shown as supplementary Table 1 (this is a request by another reviewer). We agree that the concept of Figure 5 may be suitable for two-way ANOVA, but we did not set a Wistar rat group reared in enriched environment because that group was not necessary for this study. Then, it is difficult to employ two-way ANOVA in this Figure. As we are afraid that we may have misunderstood this comment, we would appreciate if this point regarding two-way ANOVA could be more clarified.]

  1. Fig6C represents an IHC analysis and 12E pictures have clearly visible torn areas that alone makes those pictures to be unsuitable for further analysis. These pictures should be replaced by good quality sections and counting should be re-done.

[We have added enlarged micrographs showing single and double-positive cells more clearly as a supplementary Figures. As similar request was made by another Reviewer, we have counted the number of cFos+/NeuN+ cells in presumable II/III and V/VI layers. The counting data were almost the same as the presented one as shown in the below. However, we could not purchase appropriate antibodies in time to identify the cortical layers, so we abandoned the data. We are sorry not to respond well to the suggestion.]

Round 2

Reviewer 3 Report

Utsunomiya et al provided an excellent work to significantly improve the quality of their manuscript, while some questions are still unaddressed. Overall, I suggest to accept this paper to publication as scientific soundness has been increased and added parts cover most of the concerns.